# Weight Status and Myopia in Children and Adolescents: A Nationwide Cross-Sectional Study of China

**DOI:** 10.3390/nu17020260

**Published:** 2025-01-12

**Authors:** Chunjie Yin, Qian Gan, Peipei Xu, Titi Yang, Juan Xu, Wei Cao, Hongliang Wang, Hui Pan, Ruihe Luo, Hui Xiao, Kai Wang, Qian Zhang

**Affiliations:** 1School of Public Health, Xinjiang Medical University, Urumqi 830017, China; yinchunjie2020@163.com (C.Y.); xh20108262@sina.com (H.X.); wangkaimath@sina.com (K.W.); 2Chinese Center for Disease Control and Prevention, National Institute for Nutrition and Health, Key Laboratory of Public Nutrition and Health, National Health Commission of the People’s Republic of China, Beijing 100050, China; gqian212@outlook.com (Q.G.); xupp.122@163.com (P.X.); didi2001@163.com (T.Y.); 13521627903@126.com (J.X.); caow2012@126.com (W.C.); andrea0417@126.com (H.W.); panhui@ninh.chinacdc.cn (H.P.); luorh@ninh.chinacdc.cn (R.L.)

**Keywords:** near-sightedness, obesity, body mass index, children, body fat percentage

## Abstract

Background/Objectives: The prevalence of both myopia and obesity is increasing among children and adolescents around the world. We aimed to examine the association between weight status and myopia in Chinese children and adolescents. Methods: The analysis included 35,108 participants aged 6–17 from a nationwide cross-sectional survey. Results: Our results indicated that girls with overweight and obesity had higher odds ratio (OR) of myopia and mild myopia: ages 6–8 (OR = 1.33, 95% CI: 1.01–1.76; OR = 1.71, 95% CI: 1.14–2.55) and 9–11 (OR = 1.22, 95% CI: 1.03–1.44; OR = 1.31, 95% CI: 1.06–1.62). Myopic boys had higher body fat percentages (BFP) at ages 6–8 (β = 1.18, 95% CI: 0.38–1.98), 9–11 (β = 0.58, 95% CI: 0.08–1.07), and 15–17 (β = 0.42, 95% CI: 0.03–0.80), and higher body mass index (BMI) at ages 9–11 (β = 0.30, 95% CI: 0.12–0.48) and 12–14 (β = 0.19, 95% CI: 0.02–0.38). Girls had higher BFP at ages 9–11 (β = 0.62, 95% CI: 0.19–1.05) and 12–14 (β =0.53, 95% CI: 0.19–0.88) and higher BMI at 9–11 (β = 0.53, 95% CI: 0.34–0.73). Furthermore, the dose–response curves revealed a significant gender- and age-specific association between BMI, BFP, and myopia odds ratio. Conclusions: the study found an association between overweight or obesity and myopia in girls aged 6–11, and indicated that high BFP and BMI are associated with myopia, varying by sex and age.

## 1. Introduction

Both myopia and obesity are becoming more common among children and adolescents around the world. According to research, myopia affected about 22.9% of the world’s population in 2020, expected to rise to about 49.8% by 2050 [1]. Worldwide age-standardized prevalence of obesity increased at an alarming rate in ages 5–19 years, from 0.7% to 5.6% in girls and from 0.9% to 7.8% in boys between 1975 and 2016 [2]. In China, the prevalence of people with comorbid overweight, obesity, and myopia increased from 11.1% in 2019 to 17.9% in 2020, and the incidence of people with comorbid overweight, obesity, and myopia was 10.9% [3]. Along with the increase in people with myopia and obesity, the associated medical costs have increased substantially and constantly. A recent estimate shows that myopia-related visual impairment caused a global potential productivity loss of around USD 250 billion in 2015 [4]. The annual medical costs for children with overweight or obesity were USD 237.55 per capita compared to healthy-weight children [5]. Consequently, myopia and overweight/obesity have emerged as the two main global public health concerns.

Currently, a nationwide study of US adolescents found that both low BMI and high BMI are associated with mild-to-moderate (a spherical equivalent of ≤−0.75 diopters to >−6.00 diopters) and severe myopia (a spherical equivalent of ≤−6.00 diopters) [6]. A population-based prospective cohort study found that children with myopia had a higher body mass index (BMI) in the Netherlands [7]. In Korea, an association exists between obesity and high myopia in childhood and adolescence [8]. Conversely, some studies also demonstrated that myopia was not associated with measurements of body stature (height, weight, and BMI) [9,10]. However, in light of a lack of nationwide data from China on the relationship between weight status and myopia, the present study aimed to examine the association between weight status and myopia based on the Survey and Application of the Nutrition and Health System for Children Aged 0–18 Years in China (2019–2021) data [11], providing scientific evidence and foundational data support for the formulation of comorbidity prevention and control policies.

## 2. Methods

### 2.1. Participants

The study analyzed data obtained between 2019 and 2021 from the National Nutrition and Health Systematic Survey and Application for 0–18-year-old children. A multistage stratified random sampling strategy was used to recruit children from 28 counties in 14 provinces in seven regions of China. Within each surveyed county, there was a consistent annual population of 196 students aged 6 to 17, with an equal distribution of boys and girls. In total, 35,108 were finally included in the analysis after all related databases were consolidated and cleaned. The Institute of Nutrition and Health Ethics Committee of the Chinese Center for Disease Control and Prevention (Grant No. 2019-011) approved this study, and all parents of the questioned children provided their informed consent by signing a consent form.

### 2.2. Physical Examination

Trained and qualified investigators uniformly measured the height (in centimeters) and weight (in kilograms) of the children. The weights of all the children were measured in the morning on an empty stomach using a digital scale (GMCS-I electronic scale; Jianmin, Beijing, China) with a precision of 100 grams. The heights of all children were determined using a column stadiometer (Jianmin, Beijing, China), with the smallest division of 0.1cm. The body mass index (BMI), expressed in kg/m^2^, is determined by dividing an individual’s weight by the square of their height. Considering the reliability of the measurements, the analysis did not include children whose BMI was either greater than 35 or less than 10.

Weight status was classified into three categories: stunting and wasting, normal weight, and overweight and obesity. The stunting and wasting classification was based on health industry standards, such as the Screening Standard for Malnutrition of School-age Children and Adolescents (WS/T 456-2014) [12] in China. The classification of overweight and obesity was based on health industry standards, such as the Screening for Overweight and Obesity among School-age Children and Adolescents (WS/T 586-2018) [13] in China. This standard applies to school-age children and adolescents from all regions of China with different socio-economic backgrounds (including ethnic minorities).

The bioelectrical impedance technique (InBody770 body composition analyzer, Korea) measured the percentage of body fat (BFP). This method is based on four principles: direct, segmental, and multi-frequency measurement with 8-point tactile electrodes (MSD-BIA) that allow the direct measurement of body composition [14]. The technique achieved results comparable to the reference method (DXA) [15,16].

### 2.3. Questionnaire Survey

The questionnaire collected the general characteristics and myopia status of study participants’ data, including date of birth, sex, type of survey site, and myopia status. The information regarding the status of myopia was obtained by asking a series of questions concerning myopia or diopters. Since the correlation coefficient between the spherical equivalent of the right and left eyes was high (r = 0.93, *p* < 10^−300^), only data from the right eye were used for homogeneous comparison [17]. According to the Application Standard for the Detection and Prevention of Myopia in Children and Adolescents (2018) [18], myopia is a condition in which the degree of refraction in the right eye is less than −0.5 diopters (D). In addition, myopia was categorized into two groups: mild myopia (more than −3.00 D and less than or equal to −0.5 D) and moderate-to-high myopia (less than or equal to −3.00 D).

The physical activity questionnaire recorded weekly moderate-to-vigorous physical activity duration, screen time, and average daily sleep length. More specifically, moderate-to-vigorous physical activity time included the frequency and duration of motor behavior, such as transportation mode, physical education class, exercise, housework, exercise, and the intensity of each exercise. Daily screen time data encompassed the cumulative duration spent viewing television and utilizing mobile phones, laptops, and tablet devices. The average daily sleep duration was computed by summing the sleep durations on weekdays and weekends, and then dividing the total by 7. Dietary data, comprising sugar-sweetened foods and beverages, was evaluated through a food frequency questionnaire (FFQ) to determine the average intake frequency and quantity over the past month while adjusting for energy intake. More specifically, the sugar-sweetened beverage intake refers to the consumption of liquid beverages that contain added sugars, which include regular carbonated soft drinks, fruit drinks, sports drinks, energy drinks, sweetened water, and coffee or tea with added sugars. The sugar-sweetened food intake (g/d) refers to the quantity of food items that contain added sugars consumed by an individual, measured in grams per day (g/d), which include pastries (cakes/cookies/egg yolk pies/shortbread/twists, etc.), chocolate, other candies, and ice cream/popsicles/frozen treats. These questionnaires were administered by well-trained researchers and subsequently completed by the children independently.

### 2.4. Statistical Analysis

In this study, data cleaning and statistical analyses were conducted using SAS version 9.4 software (SAS Institute Inc., Cary, NC, USA), whereas R version 4.3.4 was utilized for plot drawing. Because of missing refractive data for some myopic children, the study separately analyzed myopia and its severity. Subsequently, we conducted ordinal logistic models to compare no myopia versus myopia, no myopia versus mild myopia, and no myopia versus moderate-to-high myopia. To further assess the association between weight status and myopia, we conducted a multiple linear regression analysis to analyze the association between BFP and BMI with myopia. In this study focusing on children and adolescents, a statistically significant interaction was observed between myopia and weight status. Therefore, the relationship between myopia and weight status was analyzed separately for boys, girls, and different age groups. Fully multinomial models were adjusted for residence, region of origin, screen time, sleep duration, sugar-sweetened beverage intake, and sugar-sweetened food intake. To evaluate the dose–response relationship between continuous changes in BMI and BFP and the strength of their association with myopia, we used restricted cubic splines to flexibly model the connection. Based on the Akaike information criterion (AIC), this study found that modeling flexibly with three knots at the 10th, 50th, and 90th centiles provided the best fit. All statistical tests performed during the study were two-tailed, with a significance threshold set at *p* < 0.05.

## 3. Result

### 3.1. Baseline Characteristics of the Study Participants

A total of 35,108 children were included in the study. Children’s ages ranged from 6 to 17 years old and were evenly distributed among boys and girls. Among them, 21,941 were non-myopic, and 13,140 were myopic. In total, 2379 (6.78%) children were in the stunting and wasting category, and 9822 (27.98%) in the overweight and obesity category. More detailed results about residency, region, screen time, sleep duration, sugar-sweetened food intake, sugar-sweetened beverage intake, and weight status are presented in Table 1.

### 3.2. Distribution of Overweight and Obesity, and Myopia in Children and Adolescents

In children and adolescents, the proportion of comorbid overweight/obesity and myopia, as well as the proportion of only myopia, rises with age group. In contrast, the proportion of only overweight/obesity and none of these disorders decreases with age group, as shown in Figure 1a. The proportion of girls with only myopia is higher than boys, while the proportion of boys with only overweight/obesity is higher than that of girls. The proportion of boys with comorbid overweight/obesity and myopia is higher than that of girls, as shown in Figure 1b.

### 3.3. Univariate Analysis of Myopia, Mild Myopia, and Moderate-to-High Myopia Prevalence by Weight Status, Sex, and Age Group

In a univariate logistic regression model, several sex and age group factors were found to be associated with myopia, varying by weight status. The prevalence of myopia was characterized by a significant difference in different weight statuses among boys aged 9–11 years old, namely, 14.63% in stunting and wasting and 21.28% in overweight and obesity (*p* = 0.014). However, there was no significant difference between the weight statuses of other age groups for either mild myopia or moderate-to-high myopia. For girls aged 6–8 years old and aged 9–11 years old, weight status was significantly different in myopia (*p* = 0.003 and *p* = 0.004, respectively) and mild myopia (*p* < 0.001 and *p* = 0.002, respectively) but not in moderate-to-high myopia (Table 2).

### 3.4. Risk Analysis for Myopia According to Weight Status Using Logistic Regression

Fully adjusted multinomial models were used to analyze myopia per weight status after adjusting for age, residency, region, screen time, sleep duration, sugar-sweetened beverage intake, and sugar-sweetened food intake, as shown in Figure 2 for boys and girls. A correlation matrix was applied to assess all explanatory variables for collinearity, and potential interaction terms were examined. No significant interactions were identified. Compared to normal weight status as the reference group, the adjusted odds ratio (OR) of girls with overweight and obesity was 1.33 (95% CI: 1.01–1.76, *p* = 0.041) for myopia and 1.71 (95% CI: 1.14–2.55, *p* = 0.009) for mild myopia in the 6–8 years age group, and 1.22 (95% CI: 1.03–1.44, *p* = 0.024) for myopia and 1.31 (95% CI: 1.06–1.62, *p* = 0.011) for mild myopia in the 9–11 years age group. Other age groups did not correlate with myopia in their logistic regression models, either in boys or girls.

### 3.5. Association of Myopia with BFP and BMI Among Children and Adolescents According to Sex and Age Categories

In multiple linear regression models adjusted for residency, region, screen time, sleep duration, weight status, moderate-to-vigorous physical activity time, sugar-sweetened beverage intake, and sugar-sweetened food intake, a correlation matrix was applied to assess all explanatory variables for collinearity, and potential interaction terms were examined. Compared with children and adolescents with no myopia, boys with myopia aged 6–8 years old (β = 1.18, 95% CI: 0.38–1.98, *p* = 0.004), 9–11 years old (β = 0.58, 95% CI: 0.08–1.07, *p* = 0.022), and 15–17 years old (β = 0.42, 95% CI: 0.03–0.80, *p* = 0.033) had significantly greater BFP. Further, myopia was significantly associated with higher BMI in boys aged 9–11 years old (β = 0.30, 95% CI: 0.12–0.48, *p* =0.001) and 12–14 years old (β = 0.19, 95% CI: 0.02–0.38, *p* =0.030). Among girls, compared with the reference group, myopia was significantly associated with greater BFP in the 9–11 years age group (β = 0.62, 95% CI: 0.19–1.05, *p* =0.004) and the 12–14 years age group (β = 0.53, 95% CI: 0.19–0.88, *p* = 0.002). Additionally, myopia was associated with higher BMI in girls aged 9–11 years old (β = 0.53, 95% CI: 0.34–0.73, *p* < 0.001) after controlling for residency, region, screen time, sleep duration, weight status, moderate-to-vigorous physical activity time, sugar-sweetened beverage intake, and sugar-sweetened food intake (Table 3).

### 3.6. The Dose–Response Relationship Between BMI, BFP, and Myopia Odds Ratio in Children and Adolescents by Sex and Age Group

In Figure 3, we used restricted cubic splines to visualize the dose–response relationship between BMI, BFP, and myopia odds ratio based on the binary logistic regression model adjusted for residency, region, screen time, sleep duration, moderate-to-vigorous physical activity time, sugar-sweetened beverage intake, and sugar-sweetened food intake. A correlation matrix was applied to assess all explanatory variables for collinearity, and potential interaction terms were examined. For both boys and girls in the categories 12 ≤ age ≤ 14 and 15 ≤ age ≤ 17, the odds ratios of myopia rapidly increased until around 22.75 kg/m^2^ of BMI and then started to decrease afterwards (P for non-linearity < 0.001). In the threshold analysis, girls with a BMI below 22.75 kg/m^2^ had an OR of myopia of 1.04 (95% CI: 1.03–1.07; *p* < 0.001). Similarly, the category 6 ≤ age ≤ 8 with a BMI below 22.75 kg/m^2^ had an OR of myopia of 1.04 (95% CI: 1.00–1.08; *p* < 0.046). The category 9 ≤ age ≤ 11 with a BMI below 22.75 kg/m^2^ had an OR of myopia of 1.05 (95% CI: 1.02–1.07, *p* < 0.001) (Table 4). However, no significant association was found between BMI and myopia when the BMI was above 22.75 kg/m^2^, either in sex or age group. Further, we observed that the odds ratios of myopia increased with rising BFP in both boys and girls aged 12–14 and 15–17.

## 4. Discussion

In the current study, girls who were in the overweight and obesity category had 1.33 and 1.22 times higher odds of myopia than those with normal weight in the 6–8 years and 9–11 years age groups, respectively. Furthermore, even in the mild myopia category, girls with overweight and obesity had 1.71 times and 1.31 times higher odds, respectively, compared to those with normal weight, in the 6–8 and 9–11 age groups. However, no relationship between overweight and obesity and myopia was demonstrated in girls of higher age groups. This indicates that children between the ages of 6 and 11 are in a crucial stage of development, are more sensitive to environmental changes, and are more susceptible to people with myopia and overweight and obesity-related risk behaviors. Interestingly, no association was found between myopia prevalence and overweight and obesity in boy subjects aged 6–17 years old. Furthermore, the dose–response curves revealed a significant gender-specific association between BMI, BFP, and myopia odds ratio. Several studies have reported that female sex was an additional risk factor for myopia [19,20,21,22]. The mechanism underlying this phenomenon remains poorly understood, and some researchers have proposed gender-specific factors, such as the age at which growth spurts occur [23]. For example, adolescent girls develop earlier than boys, and axial length will increase with the changes in physical signs and secondary sexual characteristics, thereby accelerating the development of myopia. Furthermore, girls have a higher risk of developing myopia due to differences in the shape and size of their eyes, characterized by a more pronounced corneal curvature, a greater lens curvature, a shallower anterior chamber, and a shorter axial length compared to boys [3].

A study conducted in Korea showed that girls with overweight and obesity have a higher risk of developing high myopia, with 4.23-fold and 5.04-fold higher odds, respectively [8]. A national cross-sectional study involving a sample of over 1.3 million teenagers found that women with mild obesity and overweight had higher odds ratios (ORs) of developing myopia compared to the reference group with low–normal BMI. The adjusted ORs were 1.19 and 1.38 for mild-to-moderate and high myopia, respectively [6]. The results of a study in Ireland involving children aged 6–7 and 12–13 years old showed that the prevalence of myopia was significantly linked to obesity, even after adjusting for age group and ethnicity (OR = 2.7) [24]. Moreover, a study about investigating the correlation between a new measure of obesity, the weight-adjusted waist index (WWI), and myopia reported that an increased WWI level was linked to a lower risk of myopia and high myopia in the overall sample, with gender-specific variations [25]. It is worth noting that the categorization of myopia employed in the Korean study varies from the one used in our research. Therefore, it is difficult to directly compare their findings with our study’s findings.

Owing to the complex relationship between risk factors at the individual level, a growing body of research has advocated BFP as a better indicator of people with obesity than BMI, which overlooks body composition [26]. Therefore, this research analyzed the association between BMI and myopia, as well as that between BFP and myopia. Our findings showed that age and sex differences were discovered in the association between BMI, BFP, and myopia. In parallel, it has demonstrated a positive relationship between BFP or BMI and myopia, taking into account the body composition level to demonstrate the association between weight status and myopia. Additionally, a nationwide population-based cross-sectional study of 938 Korean children aged 5 to 18 years reported that higher BMI was significantly associated with high myopia [27]. According to Dong et al., a lower BMI is associated with a higher prevalence of myopia and high myopia [28]. Similarly, a British cohort study found that childhood myopia is associated with lower BMI at pre- and post-pubertal ages, leading to more severe and early-onset myopia [29]. A pilot study from Chinese students reported that a higher BMI was associated with a higher risk of myopia, up to a threshold of 25 kg/m^2^, after which an increase in BMI was no longer associated with an increased risk of myopia [30].

The underlying mechanism relationship between people with obesity and myopia is not well known. According to a national study in China, there are genetic similarities between individuals who suffer from obesity and myopia. Evidence suggests that the polymorphism of the matrix metalloproteinase gene is linked to the susceptibility of obesity by affecting the remodeling of adipose tissue and to the risk of myopia by influencing the synthesis of scleral elastin [3]. Otherwise, it has been suggested that a diet-related condition known as chronic hyperinsulinemia may contribute to the development of juvenile-onset myopia through its interaction with the hormonal regulation of vitreal chamber growth [31]. Insulin secretion is suppressed under high blood sugar levels in those without diabetes, which worsens myopia by thickening the lens and shifting the anterior pole forward [32]. In addition, previous studies reported that people with myopia and obesity share common risk factors of lifestyle. For instance, obesity prevention guidelines recommend that children should have at least five servings of fruits and vegetables daily, limit their screen time to no more than two hours per day, engage in one hour of physical activity daily, and avoid consuming sugar-sweetened beverages [33]. Meanwhile, increased outdoor activities [34] and fruit and vegetable intake [35] and decreased screen time [36] and sugar-sweetened beverage or food intake [37,38] also offer a protective benefit against the risk of myopia. On the other hand, China’s rigorous education system forces students to spend more time completing homework and less time on physical activities, leading to a sedentary lifestyle, which is a shared risk factor for myopia and obesity [39]. Therefore, common strategies such as reducing sedentary behavior and screen time, improving sleep duration, limiting high-sugar diets, and increasing outdoor activities can be promoted to simultaneously prevent myopia and obesity, two common conditions among children and adolescents [40,41].

This research involved a substantial and inclusive sample size, including a wide geographical area and accurately representing the weight status and occurrence of myopia in children aged 6–17 years throughout various areas of China. It thus provides basic data for the establishment of the policy of the “Co-morbidity, Shared Etiology, and Shared Prevention” of myopia and obesity of Chinese children and adolescents. However, there are several constraints to promoting the results of the research. Due to the cross-sectional design of our investigation, correctly determining the causation between myopia and weight status was challenging. Furthermore, we did not involve cycloplegic refraction measurements for myopia because obtaining cycloplegic refraction in a large sample population is challenging, which may lead to myopia underestimation. A future study analysis to explore the impact of improving the shared susceptible environment of obesity and myopia on their comorbidity risk is necessary.

## 5. Conclusions

The current study revealed an association between myopia and weight status in girls aged 6–11 years old. Additionally, the findings revealed differences in the relationship between BMI, BFP, and myopia based on age and sex. Specifically, a positive association was observed between BFP or BMI and myopia. Further investigations with larger prospective studies are required to confirm the relationship between weight status and myopia.

## Figures and Tables

**Figure 1 nutrients-17-00260-f001:**
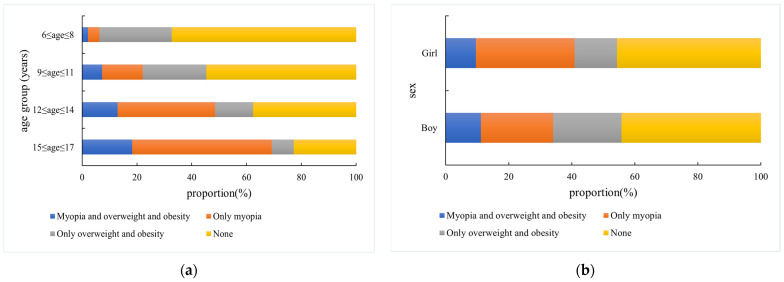
(**a**) Distribution of overweight and obesity, and myopia in children and adolescents across different age groups. (**b**) Distribution of overweight and obesity, and myopia in children and adolescents across different sexes. The bars are color-coded to represent different combinations of conditions: blue represents the proportion of children with comorbid myopia and overweight/obesity, gray represents the proportion of children with only overweight/obesity, orange represents the proportion of children with only myopia, and yellow represents the proportion of children with none of these disorders.

**Figure 2 nutrients-17-00260-f002:**
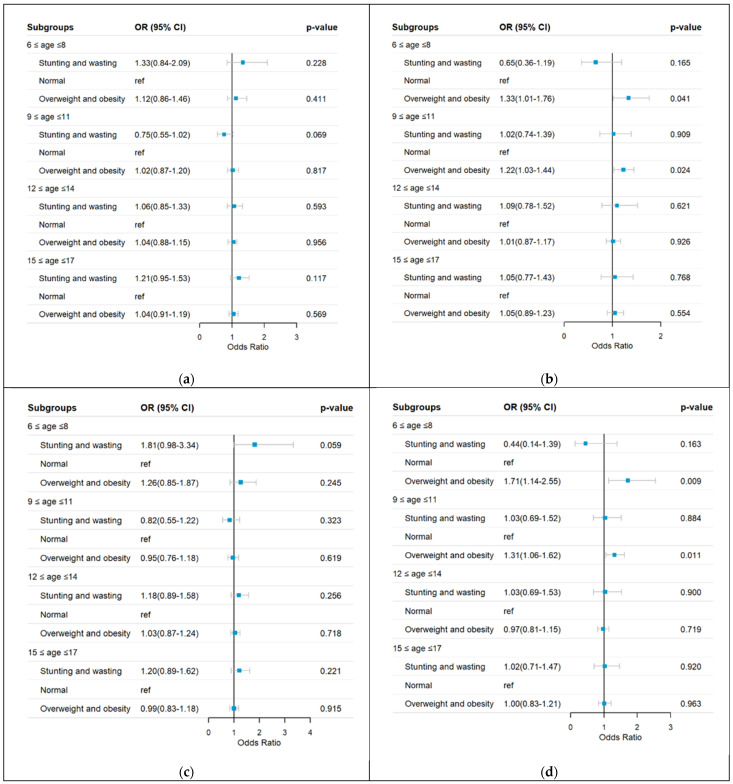
(**a**) Myopia in boys, (**b**) myopia in girls, (**c**) mild myopia in boys, (**d**) mild myopia in girls, (**e**) moderate-to-high myopia in boys, and (**f**) moderate-to-high myopia in girls. Risk analysis for the severity of myopia per weight status by sex was conducted after adjusting for residency, region, screen time, sleep duration, moderate-to-vigorous physical activity time, sugar-sweetened beverage intake, and sugar-sweetened food intake.

**Figure 3 nutrients-17-00260-f003:**
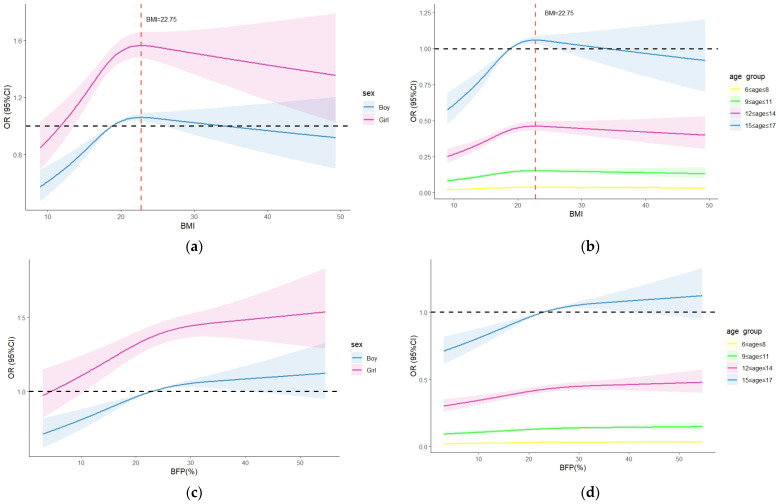
The dose–response relationship between BMI, BFP, and myopia odds ratio in children and adolescents by sex and age group. (**a**) BMI based on sex, (**b**) BMI based on age group, (**c**) BFP based on sex, (**d**) BFP based on age group. Odds ratios are indicated by solid lines and 95% CIs by shaded areas. Binary logistic regression model adjusted for residency, region, screen time, sleep duration, moderate-to-vigorous physical activity time, sugar-sweetened beverage intake, and sugar-sweetened food intake.

**Table 1 nutrients-17-00260-t001:** Baseline characteristics of the study students according to weight status.

Variable	N	Stunting and Wasting (%)	Normal (%)	Overweight and Obesity (%)
Overall	35,108	2379(6.78)	22,907 (65.25)	9822 (27.98)
Sex				
boy	17,576	1407 (8.01)	10,394 (59.14)	5775 (32.86)
girl	17,532	972 (5.54)	12,513 (71.37)	4047 (23.08)
Age group (years)				
6–8	8640	668 (7.73)	5506 (63.73)	2466 (28.54)
9–11	8236	613 (7.44)	5113 (62.08)	2510 (30.48)
12–14	8867	518 (5.84)	5970 (67.33)	2379 (26.83)
15–17	9365	580 (6.19)	6318 (67.46)	2467 (26.34)
Residency				
Urban	16,420	1139 (6.94)	10,467 (63.78)	4814 (29.32)
Rural	18,688	1248 (6.68)	12,428 (66.51)	5012 (26.82)
Region				
Northern China	17,547	1075 (6.13)	10,769 (61.37)	5703 (32.51)
Southern China	17,561	1312 (7.47)	12,129 (69.09)	4120 (23.46)
Myopia				
No	21,958	1581 (7.21)	14,195 (64.65)	6182 (28.18)
Yes	13,150	793 (6.04)	8720 (66.31)	3637 (27.68)
Moderate-to-vigorous physical activity time (min/week)				
<45	11,317	868 (7.67)	7535 (66.58)	2914 (25.75)
45–165	11,746	767 (6.53)	7651 (65.14)	3328 (28.33)
>165	12,045	744 (6.18)	7721 (64.10)	3580 (29.72)
Screen time (min/d)				
<30	9123	535 (5.86)	5993 (65.69)	2595 (28.44)
30–60	18,495	1325 (7.16)	12,047 (65.14)	5123 (27.70)
>60	7490	519 (6.93)	4867 (64.98)	2104 (28.09)
Sleep duration (h/d)				
≤9	18,864	1210 (6.41)	12,399 (65.73)	5255 (27.86)
>9	16,244	1169 (7.20)	10,508 (64.69)	4667 (28.11)
Sugar-sweetened beverage intake (g/d)				
<5.6	11,539	735 (6.37)	7213 (62.51)	3591 (31.12)
5.6–62.8	11,637	777 (6.68)	7717 (66.31)	3143 (27.01)
>62.8	11,932	867 (7.27)	7977 (66.85)	3088 (25.88)
Sugar-sweetened food intake (g/d)				
<2.4	11,603	803 (6.91)	7521 (63.82)	3280 (28.27)
2.4–13.4	11,583	781 (6.74)	7609 (65.69)	3193 (27.57)
>13.4	11,922	796 (6.68)	7777 (65.23)	3349 (28.09)

**Table 2 nutrients-17-00260-t002:** The prevalence of myopia, mild myopia, and moderate-to-high myopia by age group according to weight status.

Characteristics		Myopia n (%)	*p*-Value	Mild Myopia n (%)	*p*-Value	Moderate-to-High Myopia n (%)	*p*-Value
Boys							
6 ≤ age ≤ 8							
	Stunting and wasting	23 (7.06)	0.288	14 (4.09)	0.089	1 (0.30)	0.573
	Normal	152 (5.83)		63 (2.39)		11 (0.43)	
	Overweight and obesity	100 (7.01)		46 (3.25)		3 (0.22)	
9 ≤ age ≤ 11							
	Stunting and wasting	54 (14.63)	**0.014**	32 (9.25)	0.308	10 (3.09)	0.187
	Normal	447 (19.34)		241 (11.48)		93 (4.76)	
	Overweight and obesity	327 (21.28)		167 (12.19)		70 (5.50)	
12 ≤ age ≤ 14							
	Stunting and wasting	153 (40.05)	0.712	75 (24.67)	0.476	48 (17.33)	0.720
	Normal	1075 (40.50)		460 (22.53)		380 (19.37)	
	Overweight and obesity	590 (41.73)		263 (24.20)		196 (19.22)	
15 ≤ age ≤ 17							
	Stunting and wasting	247 (68.04)	0.231	86 (42.57)	0.610	122 (51.26)	0.182
	Normal	1887 (63.9)		681 (38.98)		932 (46.65)	
	Overweight and obesity	946 (65.42)		327 (39.54)		491 (49.55)	
Girls							
6 ≤ age ≤ 8							
	Stunting and wasting	12 (3.39)	**0.003**	3 (0.91)	**<0.001**	1 (0.29)	0.711
	Normal	174 (5.74)		68 (2.44)		19 (0.66)	
	Overweight and obesity	85 (7.98)		43 (4.50)		6 (0.61)	
9 ≤ age ≤ 11							
	Stunting and wasting	57 (21.35)	**0.004**	33 (13.75)	**0.002**	7 (3.27)	0.067
	Normal	668 (23.06)		354 (13.74)		127 (5.40)	
	Overweight and obesity	276 (27.96)		163 (18.67)		54 (7.07)	
12 ≤ age ≤ 14							
	Stunting and wasting	87 (55.41)	0.403	43 (37.72)	0.844	24 (25.26)	0.113
	Normal	1812 (53.09)		931 (36.75)		572 (26.31)	
	Overweight and obesity	545 (55.39)		268 (37.91)		192 (30.43)	
15 ≤ age ≤ 17							
	Stunting and wasting	166 (72.81)	0.533	67 (51.94)	0.849	75 (54.74)	0.334
	Normal	2560 (72.60)		1059 (52.27)		1257 (56.52)	
	Overweight and obesity	791 (74.34)		315 (53.57)		401 (59.50)	

A value of *p* < 0.05 is statistically significant and is indicated in bold font.

**Table 3 nutrients-17-00260-t003:** Association of myopia with BFP and BMI among children according to sex and age categories.

Characteristics	BFP				BMI			
	N	Mean (SD)	β (95% CI) *	*p*-Value	N	Mean (SD)	β (95% CI) *	*p*-Value
Boys								
6 ≤ age ≤ 8								
No myopia	4174	20.66 (8.84)	ref		4172	16.99 (3.21)	ref	
Myopia	282	22.61 (9.73)	1.18 (0.38–1.98)	**0.004**	282	17.41 (3.68)	0.23 (−0.07–0.52)	0.141
9 ≤ age ≤ 11								
No myopia	3367	24.48 (9.58)	ref		3363	18.89 (3.95)	ref	
Myopia	841	25.77 (9.81)	0.58 (0.08–1.07)	**0.022**	839	19.48 (4.18)	0.30 (0.12–0.48)	**0.001**
12 ≤ age ≤ 14								
No myopia	2584	20.34 (9.26)	ref		2579	20.52 (4.41)	ref	
Myopia	1875	20.27 (9.16)	−0.11 (−0.52–0.30)	0.610	1871	20.78 (4.45)	0.19 (0.02–0.38)	**0.030**
15 ≤ age ≤ 17								
No myopia	1677	19.58 (8.38)	ref		1676	22.37 (4.51)	ref	
Myopia	3122	20.05 (8.07)	0.42 (0.03–0.80)	**0.033**	3109	22.26 (4.49)	−0.12 (−0.31–0.07)	0.209
Girls								
6 ≤ age ≤ 8								
No myopia	4097	20.86 (7.86)	ref		4090	16.09 (2.75)	ref	
Myopia	277	22.85 (8.25)	0.74 (−0.01–1.49)	0.054	276	16.64 (2.95)	0.16 (−0.11–0.43)	0.237
9 ≤ age ≤ 11								
No myopia	3125	24.06 (7.85)	ref		3121	18.03 (3.65)	ref	
Myopia	1027	25.25 (8.04)	0.62 (0.19–1.05)	**0.004**	1023	18.83 (4.05)	0.53 (0.34–0.73)	**<0.001**
12 ≤ age ≤ 14								
No myopia	2071	27.51 (7.26)	ref		2067	20.21 (3.68)	ref	
Myopia	2513	28.18 (7.09)	0.53 (0.19–0.88)	**0.002**	2509	20.42 (3.64)	0.09 (−0.06–0.25)	0.242
15 ≤ age ≤ 17								
No myopia	1285	30.85 (6.61)	ref		1282	21.80 (3.72)	ref	
Myopia	3505	31.21 (6.28)	0.16 (−0.18–0.50)	0.363	3500	21.81 (3.64)	−0.08 (−0.26–0.09)	0.371

N, number; SD, standard deviation; CI, confidence interval. * Multivariate model includes adjustment for residency, region, screen time, sleep duration, weight status, moderate-to-vigorous physical activity time, sugar-sweetened beverage intake, and sugar-sweetened food intake. A value of *p* < 0.05 is statistically significant and is indicated in bold font.

**Table 4 nutrients-17-00260-t004:** Threshold effect analysis of the association between BMI and myopia by sex and age group.

Threshold of BMI	<22.75 kg/m^2^			≥22.75 kg/m^2^		
	OR	95% CI	*p*-Value	OR	95% CI	*p*-Value
Boys	1.01	0.99–1.03	0.554	0.99	0.97–1.01	0.448
Girls	1.04	1.03–1.07	**<0.001**	0.99	0.97–1.02	0.611
6 ≤ age ≤ 8	1.04	1.00–1.08	**0.0** **46**	0.98	0.90–1.08	0.755
9 ≤ age ≤ 11	1.05	1.02–1.07	**<0.001**	1.01	0.98–1.05	0.495
12 ≤ age ≤ 14	1.02	0.99–1.04	0.145	0.99	0.96–1.02	0.634
15 ≤ age ≤ 17	0.99	0.96–1.02	0.671	0.98	0.96–1.01	0.240

Model adjusted for residency, region, screen time, sleep duration, moderate-to-vigorous physical activity time, sugar-sweetened beverage intake, and sugar-sweetened food intake. A value of *p* < 0.05 is statistically significant and is indicated in bold font.

## Data Availability

The research data are not publicly available due to privacy considerations. However, interested researchers can request access to the data presented in this study by contacting the corresponding author. We will review and process data requests while ensuring privacy protection.

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
