# Peer review of "Weight Status and Myopia in Children and Adolescents: A Nationwide Cross-Sectional Study of China"

_nutrients, 2025, doi:10.3390/nu17020260_

Round 1

Reviewer 1 Report

Comments and Suggestions for Authors

1. On a global scale the topic is not new, were several papers have been published on the topic even in Asian populations (i.e. Korean). Moreover similar studies with cross-sectional and longitudinal designs has been conducted on the topic in China, and the issue has been also raised

·      Obesity and high myopia in children and adolescents: Korea National Health and Nutrition Examination Survey: https://pmc.ncbi.nlm.nih.gov/articles/PMC8956184/

·      Association between weight-adjusted waist index and myopia in adolescents and young adults: results from NHANES 1999–2008: https://bmcophthalmol.biomedcentral.com/articles/10.1186/s12886-024-03282-3

·      Inverse L-Shaped Association Between Body Mass Index and Myopia in Chinese Schoolchildren: A Pilot Study: https://www.dovepress.com/inverse-l-shaped-association-between-body-mass-index-and-myopia-in-chi-peer-reviewed-fulltext-article-JMDH

·      Myopia rate among children and adolescents amounts to 51.9% in 2022: China’s National Disease Control and Prevention Administration: https://www.globaltimes.cn/page/202403/1308787.shtml

For this reason authors should clearly and extensively indicate what is the added value of their study, in both Introduction and Discussion sections, which justify the replication

2. The title should include the design of the study, cross-sectional? Longitudinal?

3. The abstract lack for any background statements, moreover the presentation of results in the abstract is highly numeric, and should be improved and better presented. 

4. In the entire manuscript should adhere to “association” in the case of cross-sectional study rather than “link”, “correlation”, whereas they should use “relationship” in longitudinal design. This point is important to avoid any misunderstanding regarding the cause-effect interpretation

5. The included keyword should differ from those who appear in the title, to increase the chance to detect the article through a search strategy.

6. In the Introduction section, the aim of the study should be followed by a hypothesis that authors have formulated prior to the conduction of the study. What did they expect?

7. Regarding the statistical analysis, wherever there is a multiple group comparison (more than two), an adjustment of significance should be considered (i.e. Bonferroni)

8. Regarding the statistical analysis, al the models of correlation and regression should be presented in this section, especially regarding the colinearity between variables included in the same model. Moreover and in this direction the presentation of the simple and adjusted models should be clearly presented in a table. 

9. Regarding the statistical analysis, authors are requested to consider the variable BMI percentile, rather than the crude BMI values and classification of weight status is suitable in adults and not in adolescents.

10. Across the entire manuscript, authors should avoid the use of the adjective “obese” and substitute it with “individual with obesity” since the former is highly stigmatizing not anymore acceptable.

11. Regarding the Discussion section, it needs to be entirely reorganized as follow:

·      The main finding of this study, and an extensive comparison with previous studies on the same topic

·      The strengths and limitations of the study

·      The clinical implication of the study

·      The new directions of the needed future studies

Author Response

Thank you for your comments concerning our manuscript entitled “Weight Status and Myopia in Children and Adolescents: A Nationwide Study of China” (Manuscript ID: nutrients-3364436). These comments are valuable and helpful in revising and improving our thesis, as well as providing important guidance for our research. We have studied comments carefully and have made correction which we hope meet with approval. Revised portions are marked in red in the paper. The corrections in the paper and the responds to the reviewer’s and Editor’s comments are as flowing:

Responds to the reviewer’s comments:

Reviewer 1: 

Major comments

  1. On a global scale the topic is not new, were several papers have been published on the topic even in Asian populations (i.e. Korean). Moreover similar studies with cross-sectional and longitudinal designs has been conducted on the topic in China, and the issue has been also raised

Obesity and high myopia in children and adolescents: Korea National Health and Nutrition Examination Survey: https://pmc.ncbi.nlm.nih.gov/articles/PMC8956184/

Association between weight-adjusted waist index and myopia in adolescents and young adults: results from NHANES 1999–2008: https://bmcophthalmol.biomedcentral.com/articles/10.1186/s12886-024-03282-3

 Inverse L-Shaped Association Between Body Mass Index and Myopia in Chinese Schoolchildren: A Pilot Study: https://www.dovepress.com/inverse-l-shaped-association-between-body-mass-index-and-myopia-in-chi-peer-reviewed-fulltext-article-JMDH

Myopia rate among children and adolescents amounts to 51.9% in 2022: China’s National Disease Control and Prevention Administration: https://www.globaltimes.cn/page/202403/1308787.shtml

For this reason authors should clearly and extensively indicate what is the added value of their study, in both Introduction and Discussion sections, which justify the replication

Response: Thank you very much for your valuable comments. We have revised both Introduction and Discussion sections. See lines 281-284, 300-303. We appreciate your insights, which have contributed to improving the justify of the manuscript.

  1. The title should include the design of the study, cross-sectional? Longitudinal?

Response: Thank you for bringing this to our attention. We have made the necessary adjustments to the title as per your suggestion. Please refer to lines 2-3 for the revised manuscript now. Your input is greatly appreciated.

  1. The abstract lack for any background statements, moreover the presentation of results in the abstract is highly numeric, and should be improved and better presented.

Response: Thank you very much for your correction. We have added background statements to the abstract. See lines 15-16.

  1. In the entire manuscript should adhere to “association” in the case of cross-sectional study rather than “link”, “correlation”, whereas they should use “relationship” in longitudinal design. This point is important to avoid any misunderstanding regarding the cause-effect interpretation

Response: Thank you very much for your correction. We apologize for the errors in typing. After carefully checking the entire manuscript, all relevant content has been modified in the revised manuscript now. See lines 16, 26, 133, 291, 341, 343.

  1. The included keyword should differ from those who appear in the title, to increase the chance to detect the article through a search strategy.

Response: Thank you for bringing the correction to our attention. The keyword has been updated accordingly. Please refer to line 29 for the implemented modification. Your comments is greatly appreciated.

  1. In the Introduction section, the aim of the study should be followed by a hypothesis that authors have formulated prior to the conduction of the study. What did they expect?

Response: Thank you very much for your valuable suggestion. We have revised the manuscript accordingly and included the hypothesis following the aim of the study. The revisions have been highlighted in red in the revised manuscript. Please refer to lines 51-58 for the updated content.

  1. Regarding the statistical analysis, wherever there is a multiple group comparison (more than two), an adjustment of significance should be considered (i.e. Bonferroni)

Response: We appreciate your thoughtful suggestion. However, the statistical analysis referenced in our study leverages univariate and multivariate logistic regression analysis for two-group comparisons, rather than employing methods designed for multiple group comparisons, such as Analysis of Variance (ANOVA). As such, adjustments for significance levels like Bonferroni corrections are not pertinent in this context.

  1. Regarding the statistical analysis, al the models of correlation and regression should be presented in this section, especially regarding the colinearity between variables included in the same model. Moreover and in this direction the presentation of the simple and adjusted models should be clearly presented in a table.

Response: Thank you for your suggestion. Revisions are highlighted in red in the revised manuscript. See lines 169-170, 186-187, 201-205, 228-231, 244, 248-250.

  1. Regarding the statistical analysis, authors are requested to consider the variable BMI percentile, rather than the crude BMI values and classification of weight status is suitable in adults and not in adolescents.

Response: Thank you for your comments. This study weight status was based on health industry standards, which applies to school-age children and adolescents from all regions of China with different socio-economic backgrounds (including ethnic minorities). Same as BMI percentage, the weight status classification for this study was also based on their body mass index compared to others of the same age group and gender. The corresponding supplement can be found in line 87-88.

  1. Across the entire manuscript, authors should avoid the use of the adjective “obese” and substitute it with “individual with obesity” since the former is highly stigmatizing not anymore acceptable.

Response: Thank you very much for your valuable suggestion. We have carefully revised the manuscript to replace the adjective “obese” with “individual with obesity” to ensure the language is respectful and non-stigmatizing. The changes have been made on lines 18, 43, 252, and 273.

  1. Regarding the Discussion section, it needs to be entirely reorganized as follow:
  • The main finding of this study, and an extensive comparison with previous studies on the same topic
  • The strengths and limitations of the study
  • The clinical implication of the study
  • The new directions of the needed future studies

Response:Thank you for your thoughtful suggestion. we have carefully revised the manuscript to address your feedback. All modifications have been clearly highlighted in red in the revised manuscript. See lines 324-327, 330-332, 337-339. We remain committed to ensuring the quality and clarity of our manuscript and appreciate your contribution to this process.

Yours sincerely,

Chunjie Yin

Corresponding Dr. Chunjie Yin, E-mail: yinchunjie2020@163.com

Reviewer 2 Report

Comments and Suggestions for Authors Paper is Very well written and presents interesting data, that correlate obesity and ophthalmic problems. Some small issues I found in manuscript: 1. Re-check if indeed results in height of children where in accuracy 0.1 cm (1mm);  2. Screening for Overweight  and Obesity among School-aged Children and Adolescents - not in capitalics; 3. L. 96, correct  p <10-300; 4. Why children with BMI more  35 or less than 10 where not included in investigations? 5. Did authors consider to include children with high myopia (less than -6 D?); 6. Sum of residency if children is 35,095, not 35108. What could be with a  missing children? 7. What is "Sugar-sweetened beverages intake"? Authors meant liquid beverage (e.g. Cola-type or  fruit juices in mL)  or grammys of sugar that is included? 8. Sugar sweetened foods in-take (g/d). Question as above; 9. Correct typo in Fig. 7"agegroup"; 10. Instead of "cis" I advice use "dashed"; 11. What type matrix metalloproteinase gene authors meant? I mean 1, 2 , 3 or all of them? 12. Citation 29 concern very old data (50s). If possible select more up-to date; and in final Correct huge mess with literature 13. Add to disscusion: how is miopia treating in China (special relaxing to near spectacles, Atropine solutions???), how to stop/slowly miopia. Do you have in China any recomendations to slowly/stop miopia?

Author Response

Thank you for your comments concerning our manuscript entitled “Weight Status and Myopia in Children and Adolescents: A Nationwide Study of China” (Manuscript ID: nutrients-3364436). These comments are valuable and helpful in revising and improving our thesis, as well as providing important guidance for our research. We have studied comments carefully and have made correction which we hope meet with approval. Revised portions are marked in red in the paper. The corrections in the paper and the responds to the reviewer’s and Editor’s comments are as flowing:

Responds to the reviewer’s comments:

Reviewer 2

Major comments

Paper is Very well written and presents interesting data, that correlate obesity and ophthalmic problems. Some small issues I found in manuscript:

1.Re-check if indeed results in height of children where in accuracy 0.1 cm (1mm);  

Response: Thank you for your suggestion. After carefully re-checking the original data, we confirm that height was measured to the nearest 0.1 cm on a column stadiometer. Revisions are highlighted in red in the manuscript.

  1. Screening for Overweight and Obesity among School-aged Children and Adolescents - not in capitalics;

Response: Thank you for your comments. We appreciate your feedback. However, we believe that "Screening for Overweight and Obesity among School-age Children and Adolescents" is a proper noun and, therefore, should be capitalized.

3.L. 96, correct  p <10-300;

Response: Thank you very much for your careful inspection. Upon careful review and verification,it has been confirmed that the writing in the manuscript is accurate as presented. Your attention to detail is greatly appreciated.

4.Why children with BMI more 35 or less than 10 where not included in investigations?

 Response: Thank you for your question. The exclusion of children with a Body Mass Index (BMI) greater than 35 or less than 10 from certain investigations can be attributed to measurement reliability. In children, BMI does not consistently provide an accurate measure of body fat, especially at extreme values. For individuals with BMIs over 35, the relationship between BMI and body fat becomes more complex, making it less reliable for assessing obesity's health implications. Similarly, BMI values less than 10 can indicate underweight or nutritional issues which complicate the interpretation of obesity interventions. Studies often aim for populations with a clear, standardized measurement range, which excludes these extremes to maintain validity in findings. Revisions are highlighted in red in the manuscript. See line 78-79.

5.Did authors consider to include children with high myopia (less than -6 D?);

Response: Thank you for your comments. This study have included children with high myopia. See line 104.

  1. Sum of residency if children is 35,095, not 35108. What could be with a missing children?

Response: Thank you very much for your careful inspection. We apologize for the errors in data. After carefully checking the original data, we did find that residency, region, and myopia status are missing children, which use the plural to fill in missing values from the revised manuscript now. Revisions are highlighted in red in the table1. We appreciate your insights, which have contributed to improving the clarity and quality of the manuscript.

7.What is "Sugar-sweetened beverages intake"? Authors meant liquid beverage (e.g. Cola-type or  fruit juices in mL)  or grammys of sugar that is included?

Response: Thank you for your question. In this study, "Sugar-sweetened beverage intake" refers to the consumption of liquid beverages that contain added sugars, which include regular carbonated soft drinks, fruit drinks, sports drinks, energy drinks, sweetened water, and coffee or tea with added sugars. Revisions are highlighted in red in the manuscript. See line 116-119.

8.Sugar sweetened foods in-take (g/d). Question as above;

Response: Thank you for your question. In this study, "sugar-sweetened foods intake (g/d)" refers to the quantity of food items that contain added sugars consumed by an individual, measured in grams per day (g/d), which include pastries (cakes/cookies/egg yolk pies/shortbread/twists, etc.), chocolate, other candies, ice cream/popsicles/frozen treats, etc. Revisions are highlighted in red in the manuscript. See line 119-122.

9.Correct typo in Fig. 7"agegroup";

Response: Thank you very much for your correction. We apologize for the errors in typing. All relevant content has been modified in the revised manuscript now.

10.Instead of "cis" I advice use "dashed";

Response: Thank you for your suggestion.

11.What type matrix metalloproteinase gene authors meant? I mean 1, 2 , 3 or all of them?

Response: Thank you for your comments. Evidence has shown that the polymorphism of matrix metalloproteinase (MMP) gene MMP1 to MMP10 was associated with the risk of obesity by affecting adipose tissue remodeling and the susceptibility to myopia by influencing the synthesis of scleral elastin, indicating that there are genetic similarities between myopia and obesity.

  1. Citation 29 concern very old data (50s). If possible select more up-to date; and in final Correct huge mess with literature

Response: Thank you for your suggestions. We have conducted a self-check on the presentation of the manuscript. Revisions are highlighted in red in the revised manuscript.

  1. Add to disscusion: how is miopia treating in China (special relaxing to near spectacles, Atropine solutions???), how to stop/slowly miopia. Do you have in China any recomendations to slowly/stop miopia?

Response: Thank you very much for your valuable suggestion. The modification have been clearly highlighted in red in the revised manuscript. See line 324-327.

Yours sincerely,

Chunjie Yin

Corresponding Dr. Chunjie Yin, E-mail: yinchunjie2020@163.com

Round 2

Reviewer 1 Report

Comments and Suggestions for Authors

.